# Variation of Triterpenes in Apples Stored in a Controlled Atmosphere

**DOI:** 10.3390/molecules26123639

**Published:** 2021-06-14

**Authors:** Aurita Butkeviciute, Jonas Viskelis, Mindaugas Liaudanskas, Pranas Viskelis, Ceslovas Bobinas, Valdimaras Janulis

**Affiliations:** 1Department of Pharmacognosy, Lithuanian University of Health Sciences, Sukileliu av. 13, LT-50162 Kaunas, Lithuania; Mindaugas.Liaudanskas@lsmu.lt (M.L.); Valdimaras.Janulis@lsmuni.lt (V.J.); 2Institute of Horticulture, Lithuanian Research Centre for Agriculture and Forestry, Kauno str. 30, LT-54333 Babtai, Kaunas District, Lithuania; Jonas.Viskelis@lammc.lt (J.V.); Pranas.Viskelis@lammc.lt (P.V.); Ceslovas.Bobinas@lammc.lt (C.B.)

**Keywords:** whole apple, apple peel, apple flesh, triterpenic compounds, CA, HPLC-DAD

## Abstract

Apples are seasonal fruits, and thus after harvesting apples of optimal picking maturity, it is important to prepare them properly for storage and to ensure proper storage conditions in order to minimize changes in the chemical composition and commercial quality of the apples. We studied the quantitative composition of triterpenic compounds in the whole apple, apple peel and apple flesh samples before placing them in the controlled atmosphere (CA) chambers, and at the end of the experiment, 8 months later. HPLC analysis showed that highest total amount of triterpenic compounds (1.99 ± 0.01 mg g^−1^) was found in the whole apple samples of the ‘Spartan’ cultivar stored under variant VIII (O_2_—20%, CO_2_—3%, N_2_—77%) conditions. Meanwhile, the highest amount of triterpenic compounds (11.66 ± 0.72 mg g^−1^) was determined in the apple peel samples of the ‘Auksis’ cultivar stored under variant II (O_2_—5%, CO_2_—1%, N_2_—94%) conditions. In the apple peel samples of the ‘Auksis’ cultivar stored under variant I (O_2_—21%, CO_2_—0.03%, N_2_—78.97%) conditions, the amount of individual triterpenic compounds (ursolic, oleanolic, corosolic, and betulinic acids) significantly decreased compared with amount determined before the storage. Therefore, in the apple flesh samples determined triterpenic compounds are less stable during the storage under controlled atmosphere conditions compared with triterpenic compounds determined in the whole apple and apple peel samples.

## 1. Introduction

Apples are among the most consumed fruits in the world [1], and their annual harvest worldwide reaches about 89.33 million tons [2]. Most apples are grown in China (about 44.45 million tons), the US (4.65 million tons), and in Poland (3.60 million tons) [2]. Apples are seasonal fruits, and thus after harvesting apples of optimal picking maturity [3,4], it is important to prepare them properly for storage and to ensure proper storage conditions in order to minimize changes in the chemical composition and commercial quality of the apples [5].

Scientific literature has provided data that as much as 13–38% of the harvested apples in storage decrease in quality and do not reach the end consumers [6]. During storage, it is important to minimize changes in the chemical composition, sensory and physical properties of the apples. Fruit quality is first assessed based on external quality indicators (color and size) [7,8], and only then according to other quality parameters, such as the total amount of soluble solid substances [9], starch content [10], acidity [11], and mechanical resistance (firmness and elasticity) [12]. To maintain the quality of apples and to extend their shelf life, chemical compounds such as diphenylamine, ethoxyquin, or 1-methylcyclopropene (1-MCP) are used [7,13,14]. The use of diphenylamine and its derivatives is restricted in many countries due to their potential carcinogenic effects on human health [13]. Studies have shown that 1-MCP can affect secondary metabolites that are responsible for apple aroma [15]. Fruits and vegetables have recently been increasingly stored in controlled atmosphere chambers without the use of chemical compounds [16]. Scientific literature describes controlled atmosphere storage conditions when fruits are stored in chambers with low (about 1 kPa) [11] or ultra-low (0.5 or 0.7–0.8 kPa) oxygen levels [4], high carbon levels (2–3 kPa) [17], low temperature (0.5–1.0 °C), and high relative humidity (94–96%) [18]. In fruit stored in low-oxygen and low-temperature conditions, metabolism [17], ethylene production [19], and fermentation processes [20] in the cells slow down. These factors affect the resistance of the fruit to diseases and allow for preserving the quality of the fruit and prolonging their shelf life [15].

In the healthy food chain, apples are an important source of biologically active compounds. Research has shown that apples accumulate phenolic and triterpenic compounds, sugars, and organic acids, and are rich in fiber, vitamins, and macro- and microelements [21]. Different investigations have shown that the concentration of phenolics and triterpenes in apples varies with cultivar, the part of the fruit, maturity stage, environmental conditions during growth, storage and processing conditions [22]. The composition of apple peels has gained much attention in recent years because of the metabolic changes associated with postharvest storage disorders and the dietary health benefits associated with triterpenic compounds [23]. In the apple samples, accumulated pentacyclic triterpenes have a wide range of biological activity: they reduce blood levels of low-density lipoproteins [24] and glucose [25,26], inhibit inflammatory processes [27,28,29], and have anticancer [30,31,32], antibacterial effects [33,34,35], reduce free radicals [32], inhibit ferments of tyrosinase, xanthine oxidase and urease [36], have neuroprotective [37] and cardioprotective effects [28]. Triterpenic compounds are a group of biologically active compounds with a wide range of biological activities, and their preparations can be used for the prevention of chronic diseases [25]. Considering the results of studies on the biological effects of triterpenic compounds, it is important to study the chemical composition of food products, including apples, in order to assess not only the total amount of biologically active compounds, but also to provide a detailed analysis of individual triterpenic compounds, and to create possibilities for consumers with apples and their processed products of known composition and of high quality.

Determination of the optimal storage conditions in controlled atmosphere chambers are an important technology that is innocuous and can be applied to organic fruit and vegetables. Triterpenic compounds have wide range of biological activity and are one of the groups of biologically active compounds that determine the consumption of apples in the healthy food chain. Thus, it induces to determine the whole apple, apple peel and apple flesh samples’ changes in the quantitative composition of individual triterpenic compounds during storage in different controlled atmosphere conditions. The results obtained during the study will determine the parameters under which apples stored in controlled atmosphere conditions would undergo minimal changes in the nutritional value and the quantitative composition of triterpenic compounds in various parts of apples. We put forward a hypothesis that triterpenic compounds varied in the whole apple, apple peel and apple flesh samples depending on oxygen and carbon dioxide concentrations in controlled atmospheric chambers.

The aim of the study was to determine the changes in the quantitative composition of triterpenic compounds in the whole apple, apple peel and apple flesh samples stored in controlled atmosphere chambers with different composition of the atmosphere.

## 2. Results

### 2.1. Triterpenes Variation in Whole Apple Samples before and after Storage CA

Scientific literature presents studies on the changes in the qualitative and quantitative composition as well as antioxidant activity of biologically active compounds in apple samples during storage in a controlled atmosphere or a modified atmosphere chambers [17,38]. These studies present fragmented data on the dynamics of changes in the chemical composition of triterpenic compounds in the whole apple samples during their storage. With the recent intensive development of horticulture and the production of large quantities of apples, a need has arisen to store them and extend their shelf life in order to provide the consumers with products of high quality and high nutritional value. An important group of biologically active compounds that determine the nutritional properties of apples and their effect on the prevention of various diseases are triterpenes. For this reason, we determined the changes in the quantitative composition of triterpenic compounds in the whole apple samples during storage in chambers with controlled atmosphere composition.

To begin the analysis of changes in the chemical composition of triterpenic compounds in the whole apple samples stored in controlled atmosphere chambers, we performed an analysis of the total amount of the identified triterpenic compounds. The analysis showed that before storage the total amount of triterpenic compounds in the whole apple samples varied from 1.01 ± 0.11 mg g^−1^ to 2.29 ± 0.16 mg g^−1^ (Figure 1). The highest total content of triterpenic compounds (2.29 ± 0.16 mg g^−1^) was found in the whole apple samples of the ‘Spartan’ cultivar, and the lowest content (1.01 ± 0.11 mg g^−1^)—in the samples of the ‘Cortlend’ cultivar (Figure 1). Researchers have found that the amount of triterpenic compounds in the apple samples ranged from 0.47 mg g^−1^ to 3.75 mg g^−1^ [39].

In the whole apple samples of variant VIII, the content of triterpenic compounds changed the least. The highest total amount of triterpenic compounds (1.99 ± 0.12 mg g^−1^) was found in the whole apple samples of the ‘Spartan’ cultivar stored under variant VIII conditions, and it significantly different from that found in variants I to VII, but did not differ from the total amount of triterpenic compounds found in the whole apple samples before placing them in the storage chambers (Figure 1). In the whole apple samples stored under variant V conditions, the total content of triterpenic compounds decreased by 59.92%, compared with amount determined before the storage. The evaluation showed that in the whole apple samples of the ‘Spartan’ cultivar stored under variant V conditions, the amount of triterpenic compounds dropped by 77.72%, and in the samples of ‘Auksis’ and ‘Lodel’ cultivars, it dropped by 36.43% and 37.58%, respectively compared with amount determined before the storage. Studies have shown that storage of apples under low oxygen and high carbon dioxide conditions slows down the metabolic processes in fruit cells, allowing apples to maintain a stable composition of biologically active compounds [17]. The results of our study opposed and showed that in the whole apple samples stored in 20% oxygen and 3% carbon dioxide conditions, the quantitative composition of triterpenic compounds did not change significantly. Dobrzanski et al. in their studies found that apple samples of the ‘Delicious’ cultivar should be stored under ultra-low oxygen content (0.7% O_2_ and 2% CO_2_), but storage of the apple samples of ‘Marshall’ and ‘McIntosh’ cultivars is not recommended under these conditions, as the storage of these apple samples requires a sufficiently high oxygen content (4–4.5% O_2_ and 2–3% CO_2_) [10]. Scientific literature provides data on studies on the response of apple samples of various cultivars to low-oxygen environment caused by specific metabolic processes was determined by genetic factors [11,15].

Quantitative analysis of triterpenic compounds in the whole apple samples of different cultivars revealed the presence of the following triterpenic compounds: ursolic, oleanolic, corosolic, and betulinic acids (Figure 2).

The evaluation showed that before storage in controlled atmosphere chambers, the content of ursolic acid in the whole apple samples varied from 0.77 ± 0.10 mg g^−1^ to 1.85 ± 0.17 mg g^−1^ (Figure 3). The highest content of ursolic acid (1.85 ± 0.17 mg g^−1^) was found in the whole apple samples of the ‘Spartan’ cultivar, while the lowest content (0.77 ± 0.10 mg g^−1^) was detected in the apple samples of the ‘Cortlend’ cultivar (Figure 3). Other researchers confirmed the results of our study and found that ursolic acid levels in the apple samples ranged from 0.45 mg g^−1^ to 3.52 mg g^−1^ [40], and ursolic acid was the dominant compound in the studied samples [41]. Studies have shown that ursolic acid may account for 70% or more of the total amount of triterpenic compounds found in the apple samples [23].

In the whole apple samples stored in the controlled atmosphere chambers, the quantitative composition of ursolic acid varied (Figure 3). The highest amount of ursolic acid (1.56 ± 0.01 mg g^−1^) was found in the whole apple samples of the ‘Spartan’ cultivar stored under variant VIII conditions and did not differ significantly from the ursolic acid content in the apple samples before the storage (Figure 3). The greatest decrease in ursolic acid content (62.29%) was detected in the whole apple samples stored under variant V conditions. The amount of ursolic acid in the whole apple samples of ‘Rubin’, ‘Sampion’, and ‘Spartan’ cultivars stored under variant V conditions was found to have dropped by 73.64%, 76.77%, and 80.77% respectively compared with amount of ursolic acid determined before the storage. During the storage of the apple samples, the amount of ursolic acid varied depending on the conditions of the controlled composition of gas in the chambers. The quantitative composition of ursolic acid changed the least in the whole apple samples stored under variant VIII (O_2_—20%, CO_2_—3%, N_2_—77%) conditions, while the largest decrease in the amount of ursolic acid during the storage was found in the whole apple samples stored under variant V (O_2_—5%, CO_2_—7%, N_2_—88%) conditions.

The evaluation showed that prior to the storage of apple samples in the controlled atmosphere chambers, the amount of oleanolic acid in the whole samples varied from 0.13 ± 0.08 mg g^−1^ to 0.32 ± 0.11 mg g^−1^ (Figure 4). The highest content of oleanolic acid (0.32 ± 0.11 mg g^−1^) was found in the whole apple samples of the ‘Alva’ cultivar, while the lowest content (0.13 ± 0.08 mg g^−1^) was found in the apple samples of the ‘Cortlend’ cultivar (Figure 4). The results of our studies confirm those obtained by Jemmali and Jäger, which showed that the oleanolic acid content in the apple samples varied from 0.16 mg g^−1^ to 1.0 mg g^−1^ [42,43].

The content of oleanolic acid varied the least in the whole apple samples stored under variant VIII conditions (Figure 4). The evaluation showed that the amount of oleanolic acid in the whole apple samples of ‘Auksis’ (0.19 ± 0.09 mg g^−1^), ‘Cortlend’ (0.13 ± 0.01 mg g^−1^), and ‘Sampion’ (0.27 ± 0.07 mg g^−1^) cultivars did not change significantly and did not differ from the amount of oleanolic acid found in the whole apple samples before storage (Figure 4). The study showed that the amount of oleanolic acid in the whole apple samples of the ‘Alva’ cultivar stored under variant V conditions dropped by as much as 75% compared with amount of oleanolic acid determined before the storage. The quantitative composition of oleanolic acid changed the least in the whole apple samples stored under 20% oxygen and 3% carbon dioxide conditions.

Our study showed that before storing the whole apple samples in controlled atmosphere chambers, the amount of corosolic acid in the samples ranged from 0.04 ± 0.01 mg g^−1^ to 0.20 ± 0.09 mg g^−1^ (Figure 5). The highest amount of corosolic acid (0.20 ± 0.09 mg g^−1^) was found in the whole apple samples of the ‘Rubin’ cultivar, while the lowest amount (0.04 ± 0.01 mg g^−1^) was found in the apple samples of the ‘Auksis’ cultivar (Figure 5). In scientific literature, Jäger et al. presented research data showing that the variability of the amount of corosolic acid in the apple samples may be up to 0.51 mg g^−1^ [43], and these findings confirm the results of our study.

In the whole apple samples stored under variant VIII conditions, the content of corosolic acid either did not change or changed insignificantly (Figure 5). The study revealed no statistically significant changes in the amount of corosolic acid in the whole apple samples of as many as six different cultivars—‘Alva’ (0.09 ± 0.02 mg g^−1^), ‘Auksis’ (0.04 ± 0.01 mg g^−1^), ‘Connel Red’ (0.11 ± 0.02 mg g^−1^), ‘Cortlend’ (0.11 ± 0.01 mg g^−1^), ‘Ligol’ (0.14 ± 0.03 mg g^−1^), and ‘Sampion’ (0.18 ± 0.05 mg g^−1^)—stored under variant VIII conditions (Figure 5). In the apple samples stored under variant VI conditions, the amount of corosolic decreased 48.45% compared with the amount of corosolic acid determined before the storage. Nevertheless, the quantitative composition of corosolic acid in the whole apple samples stored under the conditions of variants I to VIII conditions changed the least, and this compound remained sufficiently stable throughout the storage period.

In our study, we found that both prior to the storage in controlled atmosphere chambers and at the end of the experiment, the amount of betulinic acid in the whole apple samples was the lowest of all the identified and quantified individual triterpenic compounds. Betulinic acid was not detected in the whole apple samples stored under variants III and V–VII conditions. Betulinic acid levels were found to be unchanged or only slightly reduced in the whole apple samples stored under variants I–II and VIII conditions.

### 2.2. Triterpenes Variation in Apple Peel and Apple Flesh Samples before and after Storage CA

Scientific literature has provided data that the qualitative and quantitative composition of biologically active compounds varied between different parts of the apples i.e., it is peels and flesh [44]. In addition to phenolic compounds apple fruit contains considerable amount of lipophilic triterpenic compounds principally localized into the cuticular wax layer [45,46]. The composition of the apple peels during storage in controlled atmosphere conditions were studied, because the apple peels arerelevant in avoiding fruit damage caused by abiotic and biotic agents and help preserve the integrity and appearance for market success [23]. To evaluate the variation in the phytochemical composition of apple during the storage in controlled atmosphere conditions, the quantitative analysis of triterpenic compounds in different parts of the fruit—i.e., its peels and flesh—is appropriate. The obtained knowledge will allow for a broader use of apple peels and apple flesh as functional food and for health improvement—i.e., to produce dietary supplements, teas, and other preparations.

The analysis showed that before storage the total amount of triterpenic compounds in the apple peel samples varied from 7.69± 0.51 mg g^−1^ to 13.00 ± 0.96 mg g^−1^ (Figure 6). The highest total content of triterpenic compounds (13.00 ± 0.96 mg g^−1^) was found in the apple peel samples of the ‘Auksis’ cultivar, and the lowest content (7.69 ± 0.51 mg g^−1^)—in the samples of the ‘Rubin’ cultivar (Figure 6). Researchers have found that the amount of triterpenic compounds in apple peel samples grown in Lithuania, Poland and Estonia varied from 8.88 mg g^−1^ to 12.39 mg g^−1^ [47].

The evaluation showed that the highest total amount of triterpenic compounds (11.66 ± 0.72 mg g^−1^) was found in the apple peel samples of the ‘Auksis’ cultivar stored under variant II conditions (Figure 6). In the apple peel samples of the ‘Auksis’ cultivar stored under variant I conditions, the amount of triterpenic compounds significantly decreased, 85.78% compared with amount determined before the storage. The greatest decrease in triterpenic compounds content was detected in the apple peel samples stored under variant VIII conditions. The amount of triterpenic compounds in the apple peel samples of ‘Cortlend’, ‘Ligol’, and ‘Noris’ cultivars stored under variant VIII conditions was found to have dropped by 62.07%, 66.31%, and 61.11% respectively compared with amount determined before the storage. This study showed the opposed results of triterpenic compounds variation in the whole apple and apple peel samples. Previously discussed results found, that the highest content of triterpenic compounds was detected in the whole apple samples of variant VIII conditions, meanwhile the highest amount of triterpenic compounds in the apple peel samples was determined of variant II conditions.

In apple peel epidermal cells and the associated epicuticular wax are a rich source of secondary metabolites, including triterpenes, hydroxycinnamic acids and their fatty acid esters [22,46,48]. Corosolic, betulinic, oleanolic and ursolic acids are typical representatives of pentacyclic triterpenes, and they are widely distributed in the plant kingdom and in food products [43]. With the exception of all oleane-, ursane-, and lupane-type triterpenes, the rest were enriched within the epicuticular wax layer (>90% of total) [23]. The main triterpenic compounds detected in the apple peel sample was ursolic acid. The analysis showed that before storage the highest amount of ursolic acid (9.76 ± 0.22 mg g^−1^) was found in the apple peel samples of the ‘Auksis’ cultivar, and the lowest content (5.78 ± 0.15 mg g^−1^)—in the samples of the ‘Cortlend’ cultivar (Figure 7). Frighetto et al. have found that the amount of ursolic acid in acetone apple peel samples varied from 0.88 mg g^−1^ to 12.70 mg g^−1^ [49]. This results data confirms the of our study results.

The highest amount of ursolic acid (9.25 ± 0.18 mg g^−1^) was found in the apple peel samples of the ‘Auksis’ cultivar stored under variant II conditions and did not differ significantly from the ursolic acid content in the apple peel samples before the storage (Figure 7). In the apple peel samples of the ‘Auksis’ cultivar stored under variant I conditions, the amount of ursolic acid significantly decreased 85.60% compared with amount determined before the storage. Meanwhile, in the apple peel samples of the ‘Connel Red’ cultivar stored under variant IV conditions the amount of ursolic acid increased from 6.52 ± 0.21 mg g^−1^ to 6.75 ± 0.23 mg g^−1^ (Figure 7). Klein et al. have found that the highest amount of ursolic acid (8.92 mg g^−1^) was obtained in the apple peel sample storage in controlled atmosphere conditions compared with amount 6.60 mg g^−1^–7.09 mg g^−1^ detected in the apple peel samples storage in dynamic controlled atmosphere by respiratory quotient [50].

In the apple peel samples oleanolic acid appears with around 7% of the total wax composition [49]. The analysis showed that before storage the highest amount of oleanolic acid (2.53 ± 0.11 mg g^−1^) was found in the apple peel samples of the ‘Auksis’ cultivar, and the lowest content (1.01 ± 0.05 mg g^−1^) in the samples of the ‘Cortlend’ cultivar (Figure 8). Dashbaldan et al. obtained that the amount of oleanolic acid in the apple peel samples varied from 0.29 mg g^−1^ to 1.07 mg g^−1^, and this study approved our results [51].

The evaluation showed that the amount of oleanolic acid has not changed and low increased in the apple peel samples of the ‘Cortlend’, ‘Lodel’, and ‘Sampion’ cultivars stored under variant V conditions (Figure 8) were seen. Research evaluated that bruising increased the oleanolic acid concentration in the apple samples of the ‘Discovery’ cultivar, and the change in triterpenes concentration induced by bruising showed cultivar variation. The increased oleanolic acid concentration after bruising was most likely a fruit protection reaction induced by the mechanical injury [22]. In the apple peel samples of the ‘Auksis’ cultivar stored under variant I conditions, the amount of oleanolic acid significantly decreased 89.45% compared with amount determined before the storage (Figure 8). Klein et al. have found that the highest amount of oleanolic acid (0.62 mg g^−1^) was obtained in the apple peel sample storage in controlled atmosphere conditions compared with the amount of 0.27 mg g^−1^–0.33 mg g^−1^ detected in the apple peel samples storage in dynamic controlled atmosphere by respiratory quotient [50].

The study showed that before storage the highest amount of corosolic acid (1.31 ± 0.08 mg g^−1^) was determined in the apple peel samples of the ‘Sampion’ cultivar, and the lowest content (0.63 ± 0.07 mg g^−1^)—in the samples of the ‘Auksis’ cultivar (Figure 9). Researchers have evaluated that that the amount of corosolic acid in the apple peel samples grown in Lithuania, Poland and Estonia varied from 0.29 mg g^−1^ to 0.49 mg g^−1^ [47].

The study showed that the amount of corosolic acid has not changed and low increased in the apple peel samples of the ‘Lodel’, ‘Noris’, and ‘Spartan’ cultivars stored under variant IV conditions (Figure 9). In the apple peel samples of the ‘Lodel’ cultivar stored under variant VIII conditions, the amount of corosolic acid significantly decreased, 81.76% compared with amount determined before the storage (Figure 9). The quantitative composition of corosolic acid changed the least in the apple peel samples stored under 5% oxygen and 7% carbon dioxide conditions.

The amount of betulinic acid, compared to that of other identified and quantitatively evaluated triterpenic compounds (ursolic, oleanolic, and corosolic acids) was significantly lower. The analysis showed that before storage the highest amount of betulinic acid (0.16 ± 0.01 mg g^−1^) was found in the apple peel samples of the ‘Sampion’ cultivar (Figure 10). Dashbaldan et al. determined, that the amount of betulinic acid in the apple peel samples varied to 0.05 mg g^−1^ to 0.09 mg g^−1^ [51].

The greatest decrease in betulinic acid content (77.90%) was detected in the apple peel samples stored under variant VIII conditions (Figure 10). The amount of betulinic acid in the apple peel samples of ‘Alva’, ‘Auksis’, ‘Sampion’ and ‘Spartan’ cultivars stored under variant VIII conditions was found to have dropped by 95.97%, 93.71%, 93.41%, and 90.02% respectively compared with amount of betulinic acid determined before the storage.

Generally, the content of triterpenic compounds is particularly high in the peel compared with the flesh (the peel contains from 18.4–44.9 times more triterpenes than the flesh), and therefore the consumption of apples with peel is highly recommended [52]. The study showed that before storage the highest total amount of triterpenic compounds (0.51 ± 0.10 mg g^−1^) was determined in the apple flesh samples of the ‘Alva’ cultivar, and the lowest content (0.08 ± 0.01 mg g^−1^)—in the samples of the ‘Connel Red’ cultivar (Figure 11).

In the apple flesh samples of the ‘Alva’ cultivar stored under variant V, and VI conditions, the total amount of triterpenic compounds decreased the lowest and seek 0.44 ± 0.07 mg g^−1^ and 0.43 ± 0.11 mg g^−1^ respectively (Figure 11). The greatest decrease in total content of triterpenic compounds 65.39% and 67.41% was detected in the apple flesh samples stored under variants III and VIII conditions respectively (Figure 11). The total amount of triterpenic compounds has not detected in the apple flesh samples of ‘Ligol’, ‘Rubin’, and ‘Cotlend’ cultivars stored under variants III, IV, and VI conditions respectively (Figure 11). The analysis showed that in the apple flesh samples determined triterpenic compounds are less stable during the storage under controlled atmosphere conditions compared with triterpenic compounds detected in the whole apple and apple peel samples.

Like in the whole apple and apple peel samples, ursolic acid was the predominant compound among all the determined triterpenic compounds. The study showed that before storage the highest amount of ursolic acid (0.46 ± 0.04 mg g^−1^) was determined in the apple flesh samples of the ‘Alva’ cultivar, and the lowest content (0.06 ± 0.01 mg g^−1^)—in the samples of the ‘Connel Red’ cultivar (Figure 12). Researchers have evaluated that the greatest amount of ursolic acid (0.25 mg g^−1^) was detected in the apple flesh samples of the ‘Rajka’ cultivar [52].

The content of ursolic acid varied as like the total amount of triterpenic compounds detected in the apple flesh samples stored under controlled atmosphere conditions. In the apple flesh samples of the ‘Auksis’ cultivar stored under variants I–VIII conditions the amount of ursolic decreased the greatest (55.42–92.51%) compared with other cultivars (Figure 12). In the apple flesh samples of the ‘Alva’ cultivar stored under all variants I–VIII conditions the content of oleanolic and corosolic acids decreased the lowest compared with other cultivars. The amount of betulinic acid has not been detected in the apple flesh sample stored under variants I–VIII conditions. The study showed that the individual triterpenic compounds in the apple flesh samples stored under controlled atmosphere conditions are unstable and break down.

## 3. Discussion

The introduction of fruit and vegetables of high nutritional value into the market is a multi-stage process involving the identification of indicators that define their quality. It is important to keep the apple crop grown and prepared in orchards in good commercial condition for as long as possible, with minimal change in the complex of biologically active compounds. Controlled atmospheres and initial low oxygen stress are some of the common non-chemical postharvest treatments used by the pome industry [53]. These technologies are known for reducing ethylene biosynthesis and respiration rate, the key biochemical processes during fruits storage [54]. The atmosphere composition surrounding and within the produce influence cellular metabolism, causing a reduction in catabolism in climacteric fruit and vegetables and an inhibition of enzymatic reactions. Each fruit and vegetables have its own optimal controlled atmosphere conditions which, together with controls on storage duration, relative humidity and ethylene concentration, may influence shelf-life and flavor-life [55].

We evaluated changes in the quantitative composition of triterpenic compounds in the whole apple and different parts of apple i.e., peel and flesh samples of 10 cultivars during storage in eight different controlled atmosphere chambers. The smallest changes in the total content of triterpenic compounds found in the whole apple samples of different cultivars were detected in the samples stored under variant VIII (O_2_—20%, CO_2_—3%, N_2_—77%) conditions. In the whole apple samples stored under variant V (O_2_—5%, CO_2_—7%, N_2_—88%), VI (O_2_—1%, CO_2_—3%, N_2_—94%), and VII (O_2_—10%, CO_2_—3%, N_2_—87%) conditions, the total amount of triterpenic compounds decreased by 59.92%, 53.43%, and 50.07% respectively compared with total amount of triterpenic compounds determined before the storage. The gas composition of the controlled atmosphere chambers influenced the changes in the quantitative composition of individual triterpenic compounds. The studies showed that the content of corosolic acid in the whole apple samples of six different cultivars—‘Alva’, ‘Auksis’, ‘Connel Red’, ‘Cortlend’, ‘Ligol’, and ‘Sampion’—stored in chamber VIII did not change significantly. The quantitative composition of corosolic acid in comparison with the content of ursolic, oleanolic, and betulinic acids in the whole apple samples remained sufficiently stable during storage under various conditions of gas composition in the chambers. Changes in fruit storage conditions after harvest cause abiotic stress, which causes the fruit to begin to synthesize and accumulate secondary metabolites [56,57]. The amount of biological active compounds found in fruit and vegetables has been found to depend on the composition of the atmosphere, in particular the concentration of carbon dioxide. The high levels of CO_2_ can inhibit the synthesis of biological active compounds [58]. On the other hand, it has been shown that increased CO_2_ concentrations in the atmosphere can be a factor in abiotic stress, while influencing the synthesis and quantitative composition of secondary metabolites [59].

The study showed that the quantitative composition of biologically active compounds varied between different parts of the apples i.e., peels and flesh [44]. During storage, the wax layer as a physical barrier is especially important in limiting water and weight loss in apples [60]. Previous studies have found changes in the composition of wax during apple storage and decreases in the concentrations of some ester fractions during long-term controlled atmosphere cold storage [60]. Therefore, storage conditions may play an important role for the content of bioactive compounds in apples. It has been suggested that fruits should be consumed soon after purchase and with their peel intact, as domestic storage and peeling can decrease the bioactive compound content [61]. This study showed the opposed results of triterpenic compounds variation in the whole apple and apple peel samples. Previously discussed results found, that the highest content of triterpenic compounds was detected in the whole apple samples of variant VIII (O_2_—20%, CO_2_—3%, N_2_—77%) conditions, meanwhile the highest amount of triterpenic compounds in the apple peel samples was determined of variant II (O_2_—5%, CO_2_—1%, N_2_—94%) conditions. In the apple peel samples of the ‘Auksis’ cultivar stored under variant I (O_2_—21%, CO_2_—0.03%, N_2_—78.97%) conditions, the amount of individual triterpenic compounds (ursolic, oleanolic, corosolic, and betulinic acids) significantly decreased compared with amount determined before the storage. One study on apples showed that long-term storage, both regular atmosphere cold storage and controlled atmosphere cold storage, did not influence flavonoid concentrations [62]. A study on ‘King Jonagold’ apple showed that total phenolics increased in the first three months of storage, followed by a decrease in the next six months of storage under all storage conditions tested (regular and controlled atmospheres cold storage, and pre-treatment with 1-MCP plus controlled atmosphere cold storage) [61]. During postharvest development, and fruit storage, water transport occurs through epidermal transpiration respiration, and diffusion through the peel cuticle of the fruit, a process that continues after harvest and that is one of the most important causes of mass loss during the storage of apples. In the course of this process, changes in the thickness of the cuticle, the chemical composition and amount of cuticular components [4] occur. In the apple flesh samples determined triterpenic compounds are less stable during the storage under controlled atmosphere conditions compared with triterpenic compounds determined in the whole apple and apple peel samples.

During study we identified storage parameters (gas composition, temperature, and relative humidity) that allowed for minimizing changes in the amount of individual triterpenic compounds in the whole apples and different parts of apple (peel, flesh) stored under controlled atmosphere conditions and allowed for providing the consumers with apples or their processed products containing a complex of biologically active compounds and characterized by a known composition and a high nutritional value.

## 4. Materials and Methods

### 4.1. Plant Materials

In this study, we used 10 different apple cultivars: ‘Alva’, ‘Auksis’, ‘Connel Red’, ‘Cortlend’, ‘Ligol’, ‘Lodel’, ‘Noris’, ‘Rubin’, ‘Sampion’, and ‘Spartan’. Apple samples were prepared at the Institute of Horticulture (Babtai, Lithuania), a branch of the Lithuanian Research Center for Agriculture and Forestry (coordinates: 55°60’N, 23°48’E). The altitude of Babtai town is 57 m above sea level. Trees were trained as a slender spindle. Pest and disease management was carried out according to the rules of integrated plant protection. The experimental orchard was not irrigated. Tree fertilization was performed based on the results of soil and leaf analysis. Nitrogen was applied before flowering at the rate of 80 kg ha^−1^, and potassium was applied after the harvest at the rate of 90 kg ha^−1^. Soil conditions of the experimental orchard were the following: clay loam, pH—7.3, humus—2.8%, P_2_O_5_—255 mg kg^−1^, and K_2_O—230 mg kg^−1^. The study was conducted during 2019–2020.

### 4.2. Chemicals and Solvents

All solvents, reagents, and standards used were of analytical grade. Acetonitrile, acetone, ursolic acid, oleanolic acid, betulinic acid, and corosolic acid were obtained from Sigma-Aldrich GmbH (Bethesda, MD, USA). Purified deionized water used in the tests was prepared with the Milli-Q^®^ (Millipore, Bedford, MA, USA) water purification system.

### 4.3. Controlled Atmosphere (CA) Conditions during Storage of Apple

Apples picked from different locations of the fruit tree crown were used in the study. Apple samples were stored in eight Besseling CA Systems controlled atmosphere chambers (Besseling Group, Osterblokker, Netherlands) with different gas compositions for eight months, ensuring a constant set gas composition for all eight months. The stable gas composition was controlled, and the CO_2_ released during fruit respiration was adsorbed and maintained at a constant gas composition by the Combi analysis and adsorption system (Besseling CA Systems B.V., Oosterblokker, Netherlands) with software CMB-E-2010-v14.x-1. Different controlled concentrations of oxygen, carbon dioxide, and nitrogen, constant temperature, relative humidity, and removal of endogenous ethylene were continually maintained in the controlled atmosphere chambers to prevent further fruit ripening during the storage (Table 1). Ethylene was removed by means of a scrubber-heated catalyst system MINI ADSORBER (Besseling CA Systems B.V., Oosterblokker, Netherlands) where ethylene is oxidized to yield CO_2_ and water vapor. The composition of the controlled atmosphere in the chambers was measured every 30 min and these conditions were accordingly continuously maintained with a maximum gas composition error of 0.3%. Prior to and after the 8-month storage, the qualitative and quantitative composition of triterpene compounds was evaluated in the apple samples.

### 4.4. Preparation of Samples

The whole apple fruit slices, apple peel, and apple flesh immediately frozen in a freezer (at −35 °C) with air circulation. Subsequently, these frozen samples were lyophilized with a ZIRBUS sublimator 3 × 4 × 98 5/20 (ZIRBUS technology, Bad Grund, Germany) at a pressure of 0.01 mbar (condenser temperature: −85 °C). The lyophilized samples were ground to fine powder by using an electric mill Retsch 200 (Retsch, Haan, Germany). Loss on drying before the analysis was determined by drying the apple lyophilisate in a laboratory drying oven to complete the evaporation of water and volatile compounds (temperature: 105 °C; the difference in weight between measurements: up to 0.01 g) and by calculating the difference in raw material weight before and after drying. The data were recalculated for the absolute dry lyophilisate weight. The prepared whole apple, apple peel, and apple flesh samples were stored in dark, tightly closed glass vessels.

### 4.5. Preparation of Extracts

During the analysis, 1 g of lyophilizate powder (exact weight) was weighed, added to 10 mL of acetone, and was extracted in an ultrasonic bath Sonorex Digital 10 P (Bandelin Electronic GmbH & Co. KG, Berlin, Germany) at room temperature, and at 80 kHz frequency and 1017 W power for 10 min. The conditions of the extraction were chosen based on the results of the tests for setting the extraction conditions. The obtained extract was filtered through a paper filter, and the residue on the filter was washed with acetone in a 10 mL flask until the exact volume was reached.

### 4.6. Qualitative and Quantitative Analysis by HPLC-PDA Method

A chromatograph equipped with a PDA detector Waters 2998 (Waters, Milford, CT, USA) was used for high-performance liquid chromatography (HPLC) analysis. Chromatographic separations were carried out by using an ACE (5 μm, C18, 250 × 4.6 mm inner diameter) column. The column was operated at a constant temperature of 25 °C. The volume of the analyzed extract was 10 μL. The flow rate was 1 mL min^−1^. The mobile phase consisted of acetonitrile (solvent A) and water (solvent B). We applied isocratic elution, the eluent ratio being 88% (solvent A) and 12% (solvent B). For the quantitative analysis, the calibration curve was obtained by injecting known concentrations of different standard compounds. All the identified triterpenic compounds were quantified at 205 nm wavelength [52].

### 4.7. Statistical Analysis

The study of HPLC method data was performed by using software Microsoft Office Excel 121 (Microsoft, Redmond, WA, USA) and SPSS, version 25.0 (SPSS Inc., Chicago, IL, USA). All the results obtained during the HPLC analysis were presented as means of three consecutive test results and standard deviations. Univariate analysis of variance (ANOVA) was applied in order to determine whether the differences between the compared data were statistically significant. The hypothesis about the equality of variances was verified by applying Levine’s test. If the variances of independent variables were found to be equal, Tukey’s multiple comparison test was used. The differences were regarded as statistically significant at *p* < 0.05.

## 5. Conclusions

The gas composition (oxygen, carbon dioxide and nitrogen) of the controlled atmosphere chambers influenced the changes in the quantitative composition of individual triterpenic compounds in the whole apple and different parts of apple i.e., peel and flesh samples. The study on storage conditions in controlled atmosphere chambers showed that storage under 20% oxygen and 3% carbon dioxide conditions minimized changes in the content of triterpenic compounds of the whole apples, meanwhile storage under 5% oxygen and 1% carbon dioxide conditions minimized changes in the content of triterpenic compounds of the apple peel. In the apple flesh samples, determined triterpenic compounds are more stable storage under 5% oxygen and 7% carbon dioxide conditions. However, in the apple flesh samples, determined triterpenes are less stable during the storage under controlled atmosphere conditions compared with triterpenic compounds determined in the whole apple and apple peel samples.

## Figures and Tables

**Figure 1 molecules-26-03639-f001:**
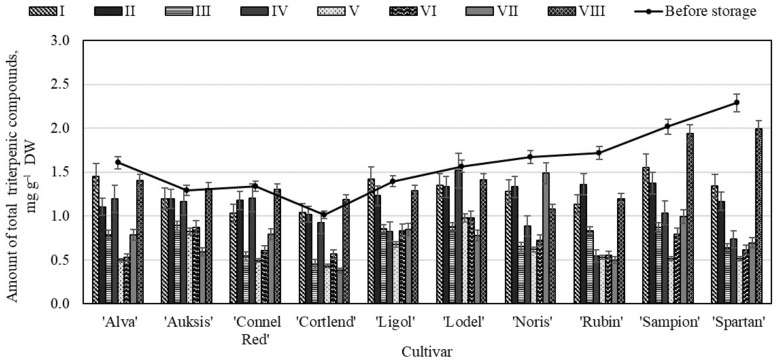
Changes in the total amount of triterpenic compounds in the whole apple samples before and after storage in CA.

**Figure 2 molecules-26-03639-f002:**
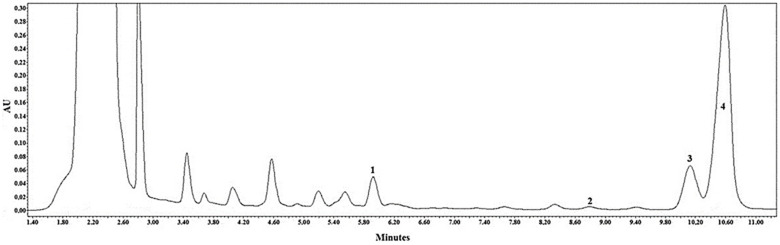
HPLC chromatogram of the apple extracts. Analytes determined at λ = 205 nm: 1—corosolic acid; 2—betulinic acid; 3—oleanolic acid; 4—ursolic acid.

**Figure 3 molecules-26-03639-f003:**
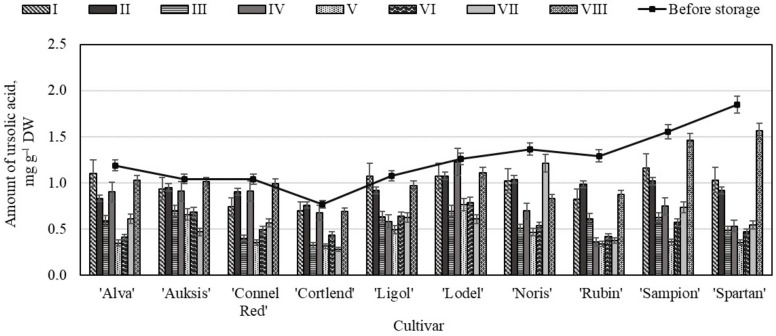
Changes in amount of ursolic acid in the whole apple samples before and after storage in CA.

**Figure 4 molecules-26-03639-f004:**
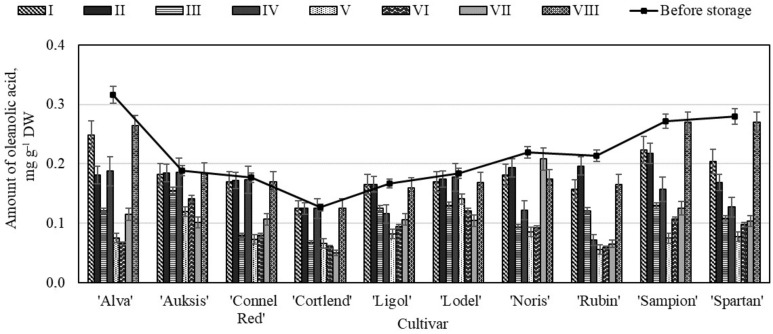
Changes in the amount of oleanolic acid in the whole apple samples before and after storage in CA.

**Figure 5 molecules-26-03639-f005:**
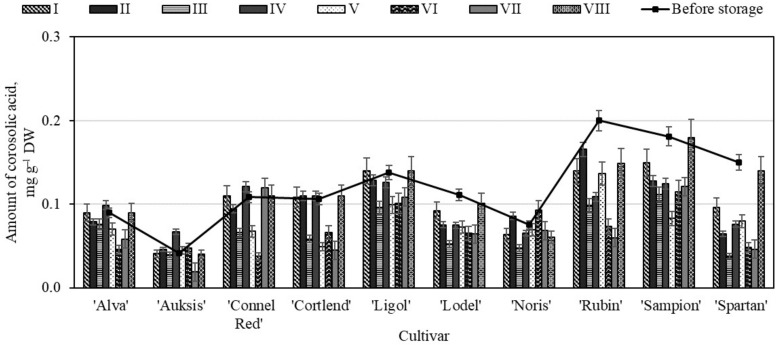
Changes in the amount of corosolic acid in the whole apple samples before and after storage in CA.

**Figure 6 molecules-26-03639-f006:**
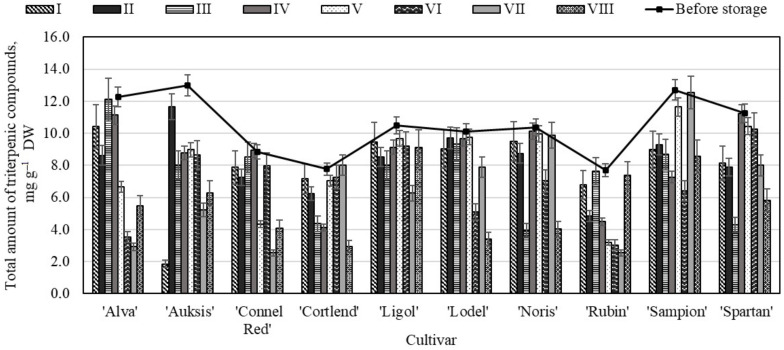
Changes in the total amount of triterpenic compounds in the apple peel samples before and after storage in CA.

**Figure 7 molecules-26-03639-f007:**
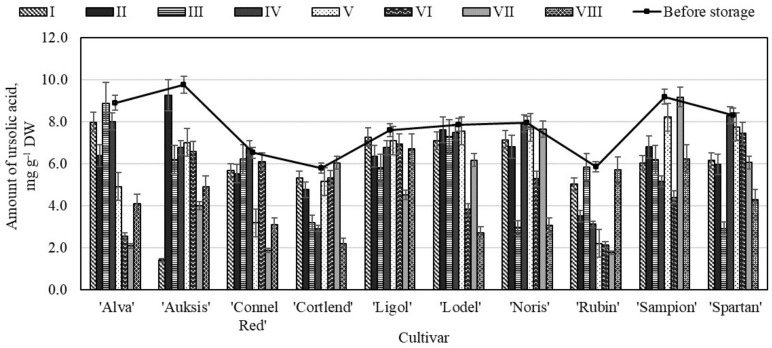
Changes in the amount of ursolic acid in the apple peel samples before and after storage in CA.

**Figure 8 molecules-26-03639-f008:**
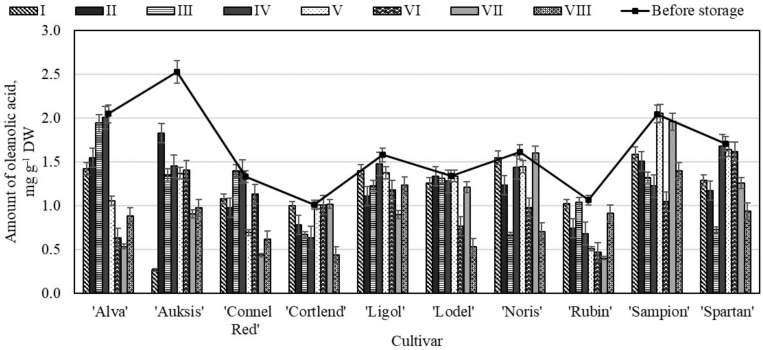
Changes in the amount of oleanolic acid in the apple peel samples before and after storage in CA.

**Figure 9 molecules-26-03639-f009:**
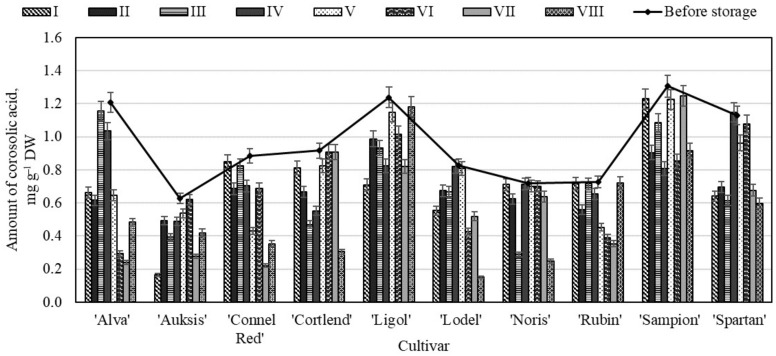
Changes in the amount of corosolic acid in the apple peel samples before and after storage in CA.

**Figure 10 molecules-26-03639-f010:**
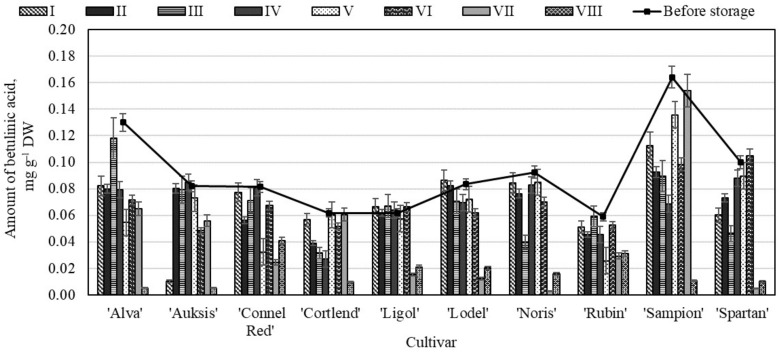
Changes in the amount of betulinic acid in the apple peel samples before and after storage in CA.

**Figure 11 molecules-26-03639-f011:**
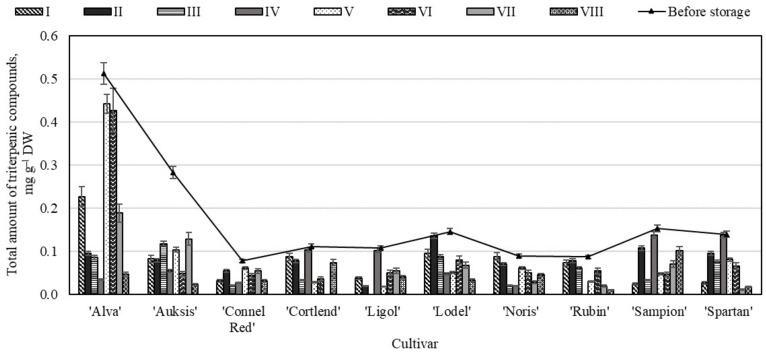
Changes in the total amount of triterpenic compounds in the apple flesh samples before and after storage in CA.

**Figure 12 molecules-26-03639-f012:**
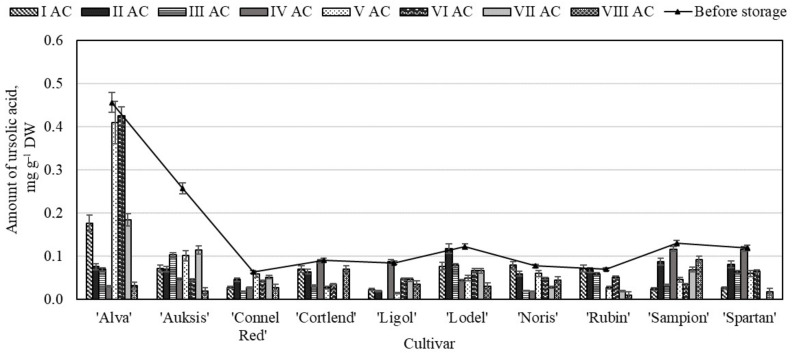
Changes in the amount of ursolic acid in the apple flesh samples before and after storage in CA.

**Table 1 molecules-26-03639-t001:** Composition of controlled atmosphere chambers.

Variant	Amount of Oxygen (O_2_), %	Amount of Carbon Dioxide (CO_2_), %	Amount of Nitrogen (N_2_), %	Relative Humidity, %	Temperature, ^o^C
I	21	0.03	78.97	95 ± 3	+1.5 ± 0.5
II	5	1	94		
III	5	3	92		
IV	5	5	90		
V	5	7	88		
VI	1	3	96		
VII	10	3	87		
VIII	20	3	77		

## Data Availability

All datasets generated for this study are included in the article.

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
