# Peer review of "Variation of Triterpenes in Apples Stored in a Controlled Atmosphere"

_molecules, 2021, doi:10.3390/molecules26123639_

Round 1

Reviewer 1 Report

This study evaluated the changes in triterpenic compounds in apples during controlled atmoshere storage for 8 months with different cultivars and different apple parts. However, the physico-chemical analysis should be determined because the apple quality is more improtant than the amount of  triterpenic compounds. In addition, more detailed discussion on the reaseon for the changes depending on storage conditions is required.

Author Response

Dear Reviewer,
thank you very much for your valuable time devoted to our manuscript and your useful comments.

Considering only the studies of the qualitative and quantitative composition of triterpene compounds, a large volume of the manuscript is formed. The analysis of physico-chemical variation of apples stored in controlled atmosphere conditions was determined, however, these data represent a large volume and so are ready new manuscript for publication. In the article, we have separated part of the discussion from the results. In the discussion part, we discussed only the changes of biologically active compounds stored in controlled atmospheric conditions. Detailed research in this field have not been found in the scientific literature, just fragmented data.

Reviewer 2 Report

Manuscript molecules-1232316 concerns an interesting issue. However, in its present form, the manuscript requires corrections and additions to improve its value. My suggestions are presented below:

  1. Preparation of Extracts: what was the power and frequency of the ultrasound during the extraction?
  2. Qualitative and Quantitative Analysis by HPLC-PDA Method: There is no information about validation of HPLC method.
  3. Exemplary chromatogram of analyzed compounds should be provided.
  4. Many items in the literature are out of date. Please, supplement the literature with new scientific reports.

Author Response

Dear Reviewer,

thank you very much for your valuable time devoted to our manuscript and your useful comments. 

1. Preparation of Extracts: what was the power and frequency of the ultrasound during the extraction?

Thank you for this comment. In 516-517 lines was supplemented: „...at room temperature, and at 80 kHz frequency and 1017 W power for 10 min“.

2. Qualitative and Quantitative Analysis by HPLC-PDA Method: There is no information about validation of HPLC method.

Thank you for this comment. Triterpenes qualitative and quantitative analysis validation by HPLC-PDA method are described in our older article (we attach the link), therefore, in order not to overload the manuscript, we quoted only the publication.

 https://www.tandfonline.com/doi/full/10.1080/10942912.2018.1506478

3. Exemplary chromatogram of analyzed compounds should be provided.

Thank you for this comment. In 152-153 lines HPLC chromatogram of the apple extracts with analytes (corosolic, betulinic, oleanolic, and ursolic acids) was provided.

4. Many items in the literature are out of date. Please, supplement the literature with new scientific reports

Thank you for this remark. References were changed, however, some literature we couldn't change, because detailed research in this field have not been found in the scientific literature, just fragmented data.

Reviewer 3 Report

Generally the manuscript with the title “Variation of Triterpenes in Apples Stored in a Controlled Atmosphere” presents a fair novelty, and the idea of the study. Since the apple is one of the most important fruit crops the present study has high scientific and practical meaning. The introduction part explains the problem and the need for current research. The methodology is well described. The experiment was well planned and prepared. The results are presented in good quality and discussed with previous reports. Conclusions are supported by the results. The manuscript can be recommended for publication in the present form.

Author Response

Thank you for the positive review.

Round 2

Reviewer 1 Report

OK. This is acceptable.